# Giant adiabatic temperature change and its direct measurement of a barocaloric effect in a charge-transfer solid

Shin-ichi Ohkoshi [1,2] ✉, Kosuke Nakagawa [1], Marie Yoshikiyo [1], Asuka Namai [1], Kenta Imoto [1], Yugo Nagane[1], Fangda Jia[1], Olaf Stefanczyk [1], Hiroko Tokoro [1,3], Junhao Wang [1,3], Takeshi Sugahara [4], Kouji Chiba[5], Kazuhiko Motodohi[6], Kazuo Isogai[6], Koki Nishioka[6], Takashi Momiki[6] & Ryu Hatano[6]

Solid refrigerants exhibiting a caloric effect upon applying external stimuli are receiving attention as one of the next-generation refrigeration technologies. Herein, we report a new inorganic refrigerant, rubidium cyano-bridged manganese–iron–cobalt ternary metal assembly (**cyano-RbMnFeCo**). **Cyano-RbMnFeCo** shows a reversible barocaloric effect with large reversible adiabatic temperature changes of 74 K (from 57 °C to –17 °C) at 340 MPa, and 85 K (from 88 °C to 3 °C) at 560 MPa. Such large reversible adiabatic temperature changes have yet to be reported among caloric effects in solid–solid phase transition refrigerants. The reversible refrigerant capacity is 26000 J kg⁻¹ and the temperature window is 142 K. Additionally, **cyano-RbMnFeCo** shows barocaloric effects even at low pressures, e.g., reversible adiabatic temperature change is 21 K at 90 MPa. Furthermore, direct measurement of the temperature change using a thermocouple shows +44 K by applying pressure. The temperature increase and decrease upon pressure application and release are repeated over 100 cycles without any degradation of the performance. This material series also possesses a high thermal conductivity value of 20.4 W m⁻¹ K⁻¹. The present barocaloric material may realize a high-efficiency solid refrigerant.

Currently, 20% of all power generated in electric power plants is used for cooling, such as air conditioners and food storage[1]. Most cooling technologies employ vapor compression cycles of gaseous refrigerants. Recently, solid refrigerants exhibiting caloric effects are receiving attention as another option. Solid-state caloric effects are caused by changes in the strength of the external field. Caloric effects caused by changes in the strength of the magnetic field, electric field, elastic force, and pressure are known as magnetocaloric[2,3], electrocaloric[4,5], elastocaloric[6–8], and barocaloric effects[9–26], respectively. The barocaloric effect has the potential to exhibit large temperature changes around room temperature[27]. Such a barocaloric effect is due to a pressure-induced change in the electronic charge state, electron spin state, or molecular orientation state in condensed matter[28]. Some solid-state condensed matter shows barocaloric effects[29].

[1]Department of Chemistry, School of Science, The University of Tokyo 7-3-1 Hongo, Bunkyo-ku, Tokyo 113-0033, Japan. [2]Cryogenic Research Center, The University of Tokyo 2-11-16 Yayoi, Bunkyo-ku, Tokyo 113-0032, Japan. [3]Department of Materials Science, Faculty of Pure and Applied Sciences, University of Tsukuba 1-1-1 Tennodai, Tsukuba, Ibaraki 305-8573, Japan. [4]Division of Chemical Engineering, Graduate School of Engineering Science, Osaka University 1-3 Machikaneyama, Toyonaka, Osaka 560-8531, Japan. [5]Material Science Div., MOLSIS Inc., 3-19-9 Hatchobori, Chuo-ku, Tokyo 104-0032, Japan. [6]Aisin Corporation, 2-1 Asahi-machi, Kariya, Aichi 448-8650, Japan. ✉e-mail: ohkoshi@chem.s.u-tokyo.ac.jp

Examples include ferroelectrics[9], perovskite-type materials[10–12], superionic conductors[13], spin-crossover materials[14,15], flexible and viscous crystals[16–20], fullerenes[21], inorganic salts[22], ferroelectric plastic crystals[23], and rubber materials[24,25]. A few practical applications of barocaloric systems, such as a cooling system designed by Barocal[30] and a cooling device using footstep pressure[31] have even been proposed. Additionally, some studies have investigated improving heat-transfer fluids to enhance the energy transfer performance of barocaloric devices[32].

For practical applications of solid-state refrigerants, the barocaloric effect in actual cooling cycles must repeatedly heat or cool its surroundings upon applying a certain pressure. There are different indicators to evaluate the performance of a reversible barocaloric effect: a reversible adiabatic temperature change ($|\Delta T_{ad,rev}|$), temperature window ($T_{span,rev}$), reversible isothermal entropy change ($\Delta S_{rev}$), and refrigerant capacity for reversible cycles ($RC_{rev}$). First-order phase transition solids, including spin-crossover materials[33–36], charge-transfer materials[37–41], and metal-insulator transition materials[42–45], may realize a giant reversible barocaloric effect due to their large entropy changes. In addition, the pre- and post-transformed phases often contribute to the barocaloric effect via a pressure-induced entropy change[5,17]. Barocaloric effects require phase transition materials that are not only highly sensitive to external pressure ($p$) but also show a large pressure-induced shift of the transition temperature (d$T$/d$p$). Consequently, cyano-bridged metal assemblies[46,47] hold promise because the structural flexibility of the −M−C≡N−M− framework should induce a large d$T$/d$p$ value.

In the present study, we report a giant reversible barocaloric effect in a rubidium cyano-bridged manganese–iron–cobalt ternary metal assembly. This assembly shows large $|\Delta T_{ad,rev}|$ values of 64 K (from 42 °C to −22 °C) at 280 MPa, 74 K (from 57 °C to −17 °C) at 340 MPa, and 85 K (from 88 °C to 3 °C) at 560 MPa. These $|\Delta T_{ad,rev}|$ values are the largest reversible adiabatic temperature changes among caloric effects in solid-solid phase transition materials reported to date. Additionally, $RC_{rev}$ and $T_{span,rev}$ also reach large values. The present material possesses $|\Delta T_{ad,rev}| = 21$ K even at 90 MPa (0.9 kbar). To confirm these behaviors, a system is constructed using a thermocouple to directly measure the temperature change ($\Delta T_{obs}$) upon applying and releasing pressure, and $\Delta T_{obs} = +44$ K (9 °C → 53 °C) is observed. In the experiments, the performance did not degrade after 10³ cycles. Furthermore, the present material exhibits a high thermal conductivity value of 20.4 W m⁻¹ K⁻¹.

## Results

### Material and crystal structure
The target material was synthesized using the following procedure. First, a mixed aqueous solution of manganese chloride and rubidium chloride was reacted with a mixed aqueous solution of potassium hexacyanoferrate, potassium hexacyanocobaltate, and rubidium chloride (Fig. 1a). Filtering and drying the precipitate yielded a powder sample (see Methods). The sample was cooled by liquid nitrogen and warmed to room temperature three times. Elemental analysis showed that the formula is RbMn{[Fe(CN)$_6$]$_{0.92}$[Co(CN)$_6$]$_{0.08}$}·0.3H$_2$O (cyano-RbMnFeCo). Scanning electron microscopy (SEM) indicated that the sample is composed of rectangular-shaped crystals with a size of $3.6 \pm 1.9$ μm (Supplementary Fig. 1). Its thermal stability was evaluated by thermogravimetry (TG) measurements in the air. The present material possesses a high heat resistivity up to 533 K (260 °C) (Supplementary Fig. 2). Thermal conductivity ($\lambda$) was measured via the thermoreflectance method using cyano-RbMnFe crystals (see Methods). The thermal conductivity displays a high value of $\lambda = 20.4 \pm 3.3$ W m⁻¹ K⁻¹.

The powder X-ray diffraction (PXRD) pattern at 300 K with Rietveld analysis indicated a cubic structure ($F\bar{4}3m$ space group) with a lattice constant of $a = 10.5589(2)$ Å (Fig. 1b, Supplementary Fig. 3

and Supplementary Table 1). The Mn site is coordinated by six N atoms of the CN ligand, and the Fe or Co site is coordinated by six C atoms of the CN ligand. A three-dimensional network is formed by cyanide bridging between Mn and Fe (or Co). Rb ions are located at every other interstitial site of the lattice, generating a non-centrosymmetric structure. For ternary metal Prussian blue analogs, mixing the transition metal ions at the C end of the CN ligand, M$_A${[M$_B$(CN)$_6$]$_x$[M$_C$(CN)$_6$]$_{1−x}$}·$z$H$_2$O, has yet to be reported, although mixing of the transition metal ions on the N end of the CN ligand (M$_{Ax}$M$_{B(1−x)}$)[M$_C$(CN)$_6$]·$z$H$_2$O is known[47].

### Temperature-induced phase transition due to charge transfer
The temperature dependence of the molar magnetic susceptibility ($\chi_M$) of cyano-RbMnFeCo was measured using a superconducting quantum interference device (SQUID) magnetometer. Figure 1c plots the product of $\chi_M$ and temperature ($T$) versus $T$. In the cooling process, the $\chi_M T$ value remains nearly constant around 4.5 K cm³ mol⁻¹ but it abruptly decreases at 192 K (≡$T_\downarrow$: temperature in the cooling process where the $\chi_M T$ change becomes half). Heating restores the original value at 248 K (≡$T_\uparrow$: temperature in the heating process where the $\chi_M T$ change becomes half). Rietveld analysis of the PXRD pattern measured at 100 K indicated a tetragonal crystal structure with lattice constants of $a = 7.1061(2)$ Å and $c = 10.5292(5)$ Å (Fig. 1d, e, Supplementary Fig. 4 and Supplementary Table 2). The observed phase transition in cyano-RbMnFeCo can be explained by a charge-transfer phase transition from the Mn$^{II}$($S = 5/2$)−NC−Fe$^{III}$($S = 1/2$) phase [high-temperature (HT) phase] to Mn$^{III}$($S = 2$)−NC−Fe$^{II}$($S = 0$) phase [low-temperature (LT) phase] as the temperature decreases. In the HT → LT phase transition, a Jahn−Teller distortion simultaneously occurs on Mn$^{III}$ of the LT phase. The symmetry is reduced from cubic to tetragonal structure due to the charge-transfer−induced Jahn−Teller (CTIJT) distortion. The electronic and spin states of the metal cations in the HT and LT phases are Mn$^{II}$($^6A_{1g}$)−NC−Fe$^{III}$($^2T_{2g}$) and Mn$^{III}$($^5B_{1g}$)−NC−Fe$^{II}$($^1A_{1g}$), respectively. Supplementary Figs. 5 and 6 show the temperature dependences of the PXRD pattern and the lattice volumes of the HT and LT phases, respectively. The volume expansion coefficients ($\beta$) for the HT and LT phases are $\beta_{HT} = 8.0 \times 10^{-6}$ K⁻¹ and $\beta_{LT} = 4.3 \times 10^{-6}$ K⁻¹, respectively.

A differential scanning calorimeter (DSC) was used to experimentally measure the transition entropy accompanying the charge-transfer phase transition of cyano-RbMnFeCo. The cooling process displays an exothermic (heat-release) peak at 196 K, while the heating process shows an endothermic (heat-storage) peak at 251 K (Fig. 1f). These values are consistent with the observed phase transition temperatures in the $\chi_M T$ versus $T$ plots. Analysis of the peak area implied that the transition enthalpy ($\Delta H_t$) values of the cooling and heating processes are 36.0 kJ kg⁻¹ and 41.7 kJ kg⁻¹, respectively. The transition entropy ($\Delta S_t$) values in the cooling and heating processes are 183 J K⁻¹ kg⁻¹ and 166 J K⁻¹ kg⁻¹, respectively.

### Pressure-induced phase transition and its reversibility
The pressure-induced effect on the charge-transfer phase transition of cyano-RbMnFeCo was investigated using a SQUID magnetometer under various applied pressures between $p = 0.1$ MPa (1 bar, ambient pressure) and 560 MPa (5.6 kbar) (see Methods). Supplementary Fig. 7 shows the $\chi_M T$−$T$ plots at various pressures. The ratio between the LT and HT phases was determined by fitting to the $\chi_M T$−$T$ plots and considering the superexchange interactions between the paramagnetic metal ions to obtain the thermal hysteresis loops in Fig. 2a and Supplementary Fig. 8 (see Methods). The $T_\downarrow$ and $T_\uparrow$ values depend on $p$: ($T_\downarrow$, $T_\uparrow$, $p$) = (197 K, 244 K, 0.1 MPa), (251 K, 334 K, 50 MPa), (267 K, 376 K, 90 MPa), (275 K, 384 K, 130 MPa), (291 K, 397 K, 170 MPa), (317 K, 410 K, 210 MPa), (327 K, 430 K, 280 MPa), (343 K, 430 K, 340 MPa), (351 K, 470 K, 380 MPa), and (375 K, 460 K, 560 MPa). Due to the upper-temperature limit of 400 K in the SQUID magnetometer, the pressure cells containing the samples at 280 MPa, 340 MPa, 380 MPa,

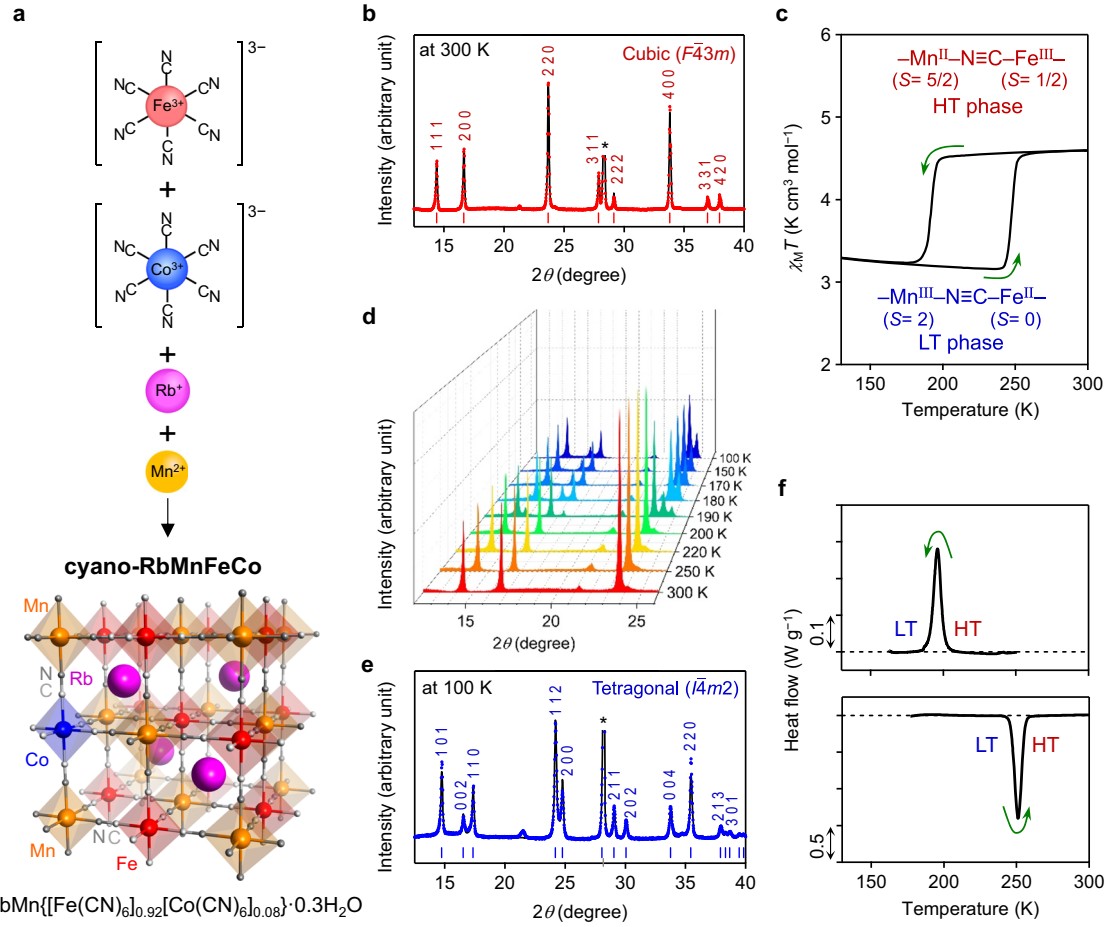

**Fig. 1 | Crystal structure and charge-transfer phase transition of cyano-RbMnFeCo. a**, Synthesis and crystal structure of RbMn{[Fe(CN)$_6$]$_{0.92}$ [Co(CN)$_6$]$_{0.08}$}·0.3H$_2$O (**cyano-RbMnFeCo**). Magenta, orange, red, blue, light gray, and gray balls represent Rb, Mn, Fe, Co, C, and N, respectively. **b**, PXRD pattern with Rietveld analysis of the HT phase at 300 K. Red dots and black line show the observed and calculated patterns, respectively. Asterisk indicates the peak from the silicon standard. **c**, Thermal hysteresis loop of cyano-RbMnFeCo measured at ambient pressure under a 5000 Oe magnetic field.

**d**, Temperature dependence of the PXRD pattern in the range of 12–26°. Front shows the initial pattern at 300 K (red) followed by PXRD patterns during the cooling process down to 100 K (blue). **e**, PXRD pattern with Rietveld analysis of the LT phase at 100 K. Blue dots and black line show the observed and calculated patterns, respectively. Asterisk indicates the peak from the silicon standard. **f**, DSC chart of cyano-RbMnFeCo depicting the cooling (upper) and heating (lower) processes.

and 560 MPa were heated up to 470 K using our lab-made heater. Then the pressure cell was immediately transferred to the SQUID magnetometer, and the measurement of the $\chi_M$ values started. The shift of the transition temperature by pressure was evaluated using the average d$T$/d$p$ value of $T_\uparrow$ and $T_\downarrow$ (Fig. 2b). The linear fitted d$T$/d$p$ value for 90 MPa and below shows a remarkably large value of 1100 K GPa$^{-1}$. This value greatly exceeds those of other barocaloric effect materials[9–26]. Upon releasing the pressure, the $\chi_M T$ value instantly recovers. The pressure-induced phase transition is reversibly observed over 20 times (Fig. 2c).

To reproduce the temperature shift of the thermal hysteresis by applying pressure, we phenomenologically analysed the present charge-transfer phase transition using the Slichter–Drickamer (SD) mean-field model[43,48]. This model describes the Gibbs free energy ($G$) of the system as $G = x\Delta H + \gamma x(1 - x) + T\{R[x\ln x + (1 - x)\ln(1 - x)] - x\Delta S\} + G_{LT}$, where $x$ is the ratio of the HT phase, $\Delta H$ is the enthalpy difference between the HT and LT phases, $\Delta S$ is the entropy difference between the HT and LT phases, $\gamma$ is the interaction parameter interpreted as the internal stress within the crystal, and $R$ is the gas constant. The $\Delta H$, $\Delta S$, and $\gamma$ values depend on the pressure. The SD model calculations well reproduce the observed shift of the thermal hysteresis at various $p$ values (Supplementary Figs. 9 and 10).

## Entropy curves under ambient and high pressures

We investigated the performance of the reversible barocaloric effect of cyano-RbMnFeCo. A physical property measurement system (PPMS) and DSC measured the heat capacity ($C_p$), while a SQUID magnetometer measured the phase transition temperatures. Using the relaxation method by PPMS, the $C_p$ versus temperature curve of the LT phase ($C_{p,LT}$) was obtained between 2 K and 240 K, whereas that of the HT phase ($C_{p,HT}$) was obtained between 174 K and 280 K (Fig. 2d). In the $C_{p,LT}$ curve, a magnetic anomaly due to the ferromagnetic phase transition occurs at the Curie temperature ($T_C$) of 11 K (Fig. 2d, inset, and Supplementary Fig. 11). $C_{p,LT}$ increases up to 240 K; then it abruptly jumps due to a phase transition to the HT phase. The entropy curve of the LT phase ($S_{LT}(T)$) was obtained by integrating $C_{p,LT}/T$ from 2 K to 240 K. Similarly, $C_{p,HT}/T$ was integrated from 196 K to 280 K to give the entropy curve of the HT phase ($S_{HT}(T)$). Based on the $\Delta S_t$ values of the phase transition from the LT to HT phase and from the HT to LT phase obtained from the DSC measurement in Fig. 1f, the average value of 175 J K$^{-1}$ kg$^{-1}$ was added to $S_{LT}(T)$ at an intermediate temperature of 224 K to accommodate for the offset of the $S_{HT}(T)$ curve (Fig. 2e). By contrast, we adopted the $S_{LT}(T, p)$ and $S_{HT}(T, p)$ curves as the pressure-dependent entropy curves of the LT and HT phases, respectively. The $S_{LT}(T, p)$ and

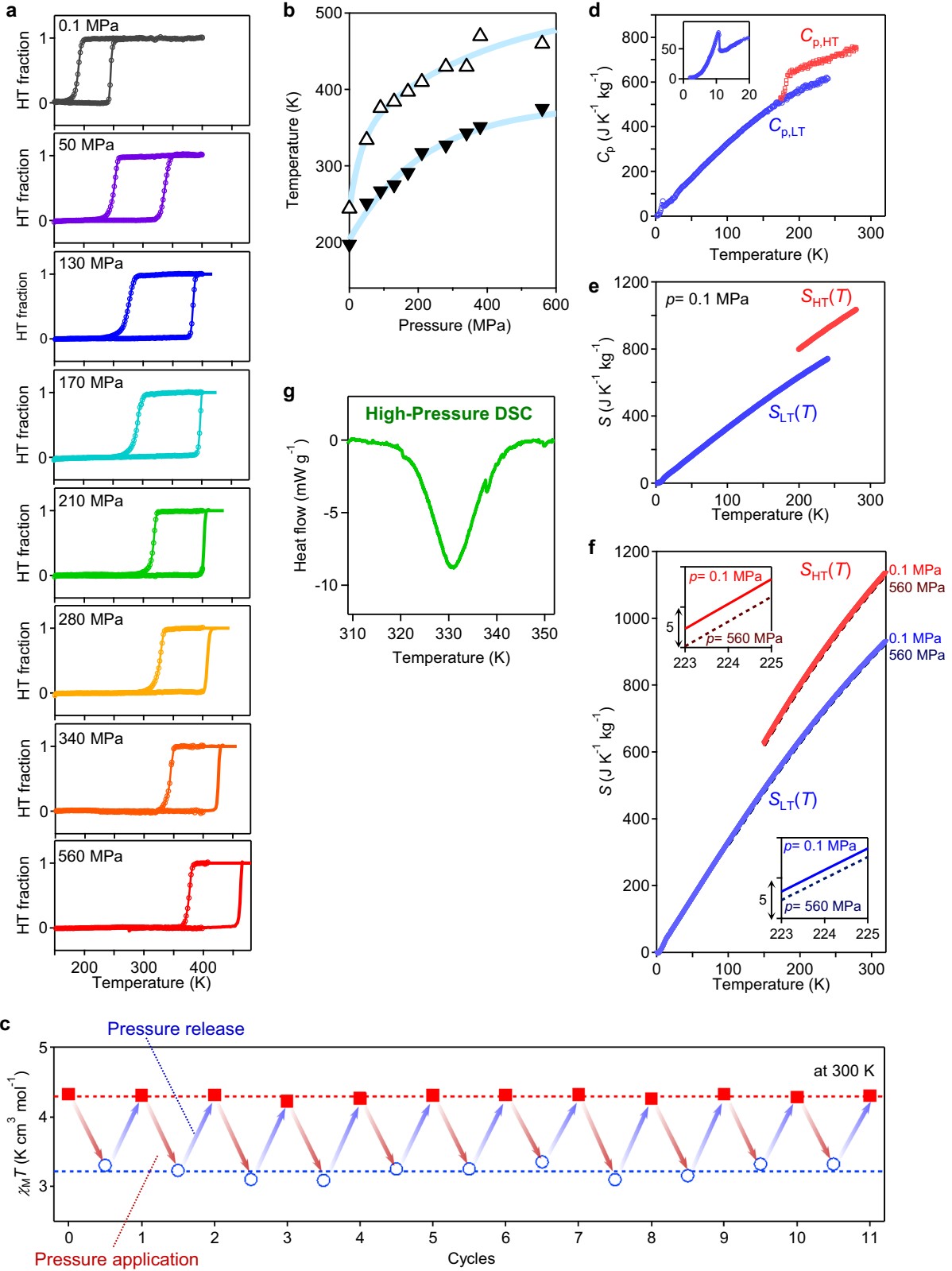

$S_{HT}(T, p)$ curves were calculated using the volume expansion coefficients and the Maxwell relation, $\sigma \equiv (\partial S/\partial p)_T = -(\partial V/\partial T)_p$. The obtained $\sigma$ values for the HT and LT phases are $\sigma = -3.9 \times 10^{-3}$ J K$^{-1}$ kg$^{-1}$ MPa$^{-1}$ and $\sigma = -1.9 \times 10^{-3}$ J K$^{-1}$ kg$^{-1}$ MPa$^{-1}$, respectively. These $\sigma$ values are very small compared to those for polymers or plastic crystals. This is reasonable because the present material is an inorganic material exhibiting zero thermal expansion. The analyses of the pressure-dependent $S_{LT}(T, p)$ and $S_{HT}(T, p)$ curves assumed that the $(\partial V/\partial T)_p$ value is independent of pressure[11,15–17,19–21]. Figure 2f shows the pressure-dependent $S_{LT}(T)$ and $S_{HT}(T)$ curves at 0.1 MPa and 560 MPa. Finally, the thermal hysteresis loops of the HT phase fraction ($x$) were calculated from the $\chi_M T-T$ plots of the SQUID measurements for each pressure and placed on the $S_{LT}(T)$ and $S_{HT}(T)$ curves using the equation $(1-x) S_{LT}(T) + x S_{HT}(T)$.

**Fig. 2 | Pressure dependence of the thermal hysteresis and its repeatability.** **a**, Thermal hystereses of the HT phase fraction versus temperature converted from the experimentally obtained $\chi_M T–T$ plots. Lines above 400 K are to guide the eye based on the HT fraction curves in the heating process. **b,** Pressure dependence of the transition temperatures of $T_\downarrow$ (filled triangles) and $T_\uparrow$ (open triangles). Light blue lines are the eye guides. **c,** Repeatability of the pressure-induced change of the $\chi_M T$ value. Red squares and blue open circles indicate the magnetic susceptibility values measured at 0.1 MPa and under pressure, respectively. Red and blue arrows indicate the process to apply and release pressure, respectively. **d**, Heat capacity ($C_p$) obtained by PPMS, **e,** entropy versus $T$ curves at $p = 0.1$ MPa, and **f,** pressure-dependent entropy versus $T$ curves for the HT (red) and LT (blue) phases. Brown and navy dotted lines denote the entropy curves at 560 MPa (5.6 kbar) for the HT and LT phases, respectively. **g**, Heat flow vs. $T$ curve of the analogous compound (see the Supplementary information) at 85 MPa (0.85 kbar) in the heating process from a high-pressure DSC measurement.

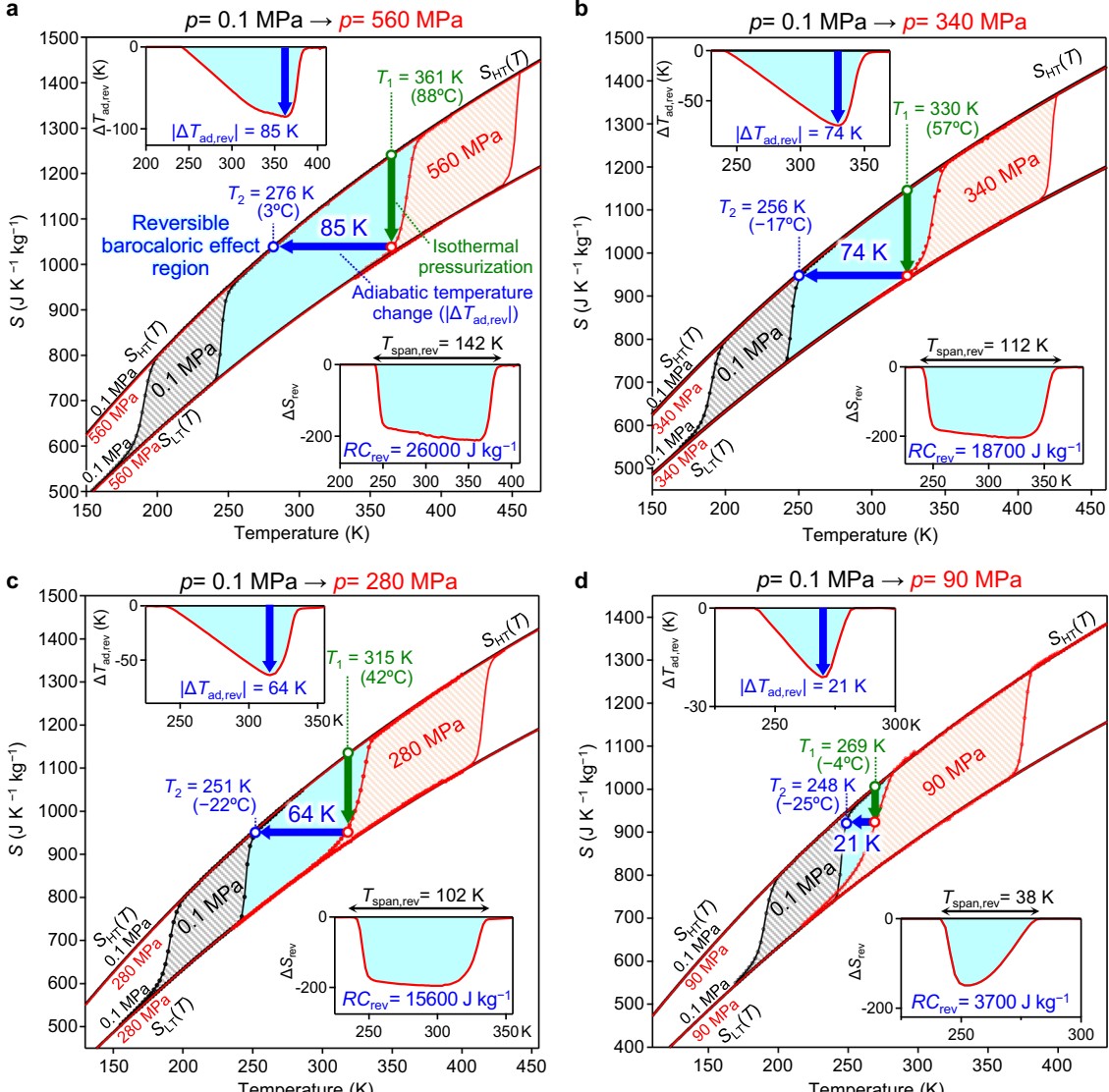

**Fig. 3 | Reversible barocaloric effect in cyano-RbMnFeCo.** Entropy versus temperature curves of the HT and LT phases of **cyano-RbMnFeCo** obtained from the heat capacity measurements using PPMS and DSC, along with the thermal hysteresis measurement using SQUID magnetometer. Black lines indicate the entropy curves at 0.1 MPa (1 bar), and red lines indicate the entropy curves at **a**, 560 MPa, **b**, 340 MPa, **c**, 280 MPa, and **d**, 90 MPa (0.9 kbar). Black and orange shaded areas indicate the thermal hysteresis loops at 0.1 MPa and at high pressures (560 MPa, 340 MPa, 280 MPa, and 90 MPa), respectively. Upper and lower insets in each figure show the temperature dependences of $\Delta T_{ad,rev}$ and $\Delta S_{rev}$, respectively. Light blue areas denote the reversible barocaloric effect regions. Green vertical arrows indicate the isothermal pressurization process, and blue thick arrows indicate the adiabatic pressure release process showing the $|\Delta T_{ad,rev}|$ values.

## Justification of the present approach for entropy curves under pressure using high-pressure DSC

Figure 2g shows the heat capacity of an analogous compound measured using a high-pressure differential scanning calorimeter (high-pressure DSC). These results demonstrate the appropriateness of the aforementioned treatment. Furthermore, since the curvature of the entropy curves at the onset and end of the phase transition under pressure is essential to evaluate $\Delta T_{ad,rev}$, we confirmed the HT phase fraction $x$ during the phase transition calculated from the $\chi_M T$ values of the SQUID data based on molecular field (MF) theory considering superexchange interactions between the paramagnetic metal cations. The $x$ values during the phase transition from SQUID data correspond to those from the high-pressure DSC measurement. (See the Supplementary Information §7 and Supplementary Figs. 12–14 for details.)

## Giant reversible adiabatic temperature change in the barocaloric effect

To evaluate the performance of the reversible barocaloric effect in the cooling cycle of cyano-RbMnFeCo, the reversible adiabatic temperature change ($\Delta T_{ad,rev}$), reversible entropy change ($\Delta S_{rev}$), temperature window ($T_{span,rev}$), and refrigerant capacity for reversible cycles ($RC_{rev}$) were calculated. The barocaloric effect involves the following process. (i) Pressure applied to the HT phase generates the LT phase via an isothermal pressurization process. This corresponds to $\Delta S_{rev}$ (green arrow in each graph of Fig. 3). (ii) Pressure is released under an adiabatic condition. This restores the HT phase at constant entropy and is accompanied by a decrease in the temperature from $T_1$ to $T_2$, corresponding to $\Delta T_{ad,rev}$ (blue arrow in each graph of Fig. 3). Afterward, (iii) the system is in contact with the surroundings. Then cyano-RbMnFeCo returns to the original HT phase through heat exchange.

The insets for each graph in Fig. 3 plot the temperature dependences of $\Delta T_{ad,rev}$ (upper left) and $\Delta S_{rev}$ (lower right). $T_{span,rev}$, which is the width of the temperature region where a reversible barocaloric effect is observed, is obtained from the width of the $\Delta T_{ad,rev}$ graph. $RC_{rev}$ is the area in the $\Delta S_{rev}$ curve (light blue area). Figure 3 shows the barocaloric effects at $p = 90$, 280, 340, and 560 MPa. As the applied pressure increases, each barocaloric parameter increases, reaching $|\Delta T_{ad,rev}| = 85$ K, $T_{span,rev} = 142$ K, $\Delta S_{rev} = -212$ J K$^{-1}$ kg$^{-1}$, and $RC_{rev} = 26000$ J kg$^{-1}$ at $p = 560$ MPa. Supplementary Fig. 15 depicts the barocaloric performances under other pressures of $p = 50$, 130, 170, 210, and 380 MPa.

The $|\Delta T_{ad,rev}|$ values of 59–85 K are the largest ones among the caloric effects in solid–solid phase transition refrigerants. For example, in the case of $|\Delta T_{ad,rev}| = 74$ K, the solid refrigerant of cyano-RbMnFeCo can cool the system from $T_1 = 330$ K (+57 °C) to $T_2 = 256$ K (−17 °C) via the barocaloric process. Supplementary Tables 3–6 compare representative barocaloric materials. Since realizing reversible barocaloric effects with low pressure is crucial, the huge values of $|\Delta T_{ad,rev}| = 21$ K, $RC_{rev} = 3700$ J kg$^{-1}$, and $T_{span,rev} = 38$ K even at 90 MPa (0.9 kbar) are attractive.

### First-principles phonon mode calculations of the reversible adiabatic temperature change

To theoretically confirm such a giant reversible barocaloric effect, we performed first-principles phonon mode calculations using MedeA Phonon code with GGA + U/PBE. In the calculations, cyano-RbMnFeCo was replaced with Rb$^I$Mn$^{III}$[Fe$^{II}$(CN)$_6$] and Rb$^I$Mn$^{II}$[Fe$^{III}$(CN)$_6$] for the LT and HT phases, respectively. The vibrational entropy $S_{vib}(T)$ curves for the LT and HT phases were obtained from the phonon mode calculations by considering the effect of thermal expansion based on the quasi-harmonic approximation method[49]. Additionally, the contributions from the orbital degeneracy and the spin multiplicity ($S_{os}$) were taken into account. The $S_{os,LT}$ value of the LT phase is $R\ln 5$ due to Fe$^{II}$($^1A_{1g}$) and Mn$^{III}$($^5B_{1g}$), whereas the $S_{os,HT}$ of the HT phase is $R\ln 36$ due to Fe$^{III}$($^2T_{2g}$) and Mn$^{II}$($^6A_{1g}$). The temperature dependences of the entropy of the LT and HT phases ($S_i(T)$, $i$ = HT or LT) at ambient pressure were obtained as $S_i(T) = S_{vib,i}(T) + S_{os,i}$ (Fig. 4a). Meanwhile, the $S_{vib,HT}(T)$ and $S_{vib,LT}(T)$ curves at $p = 280$ MPa were calculated by phonon mode calculations under pressure. Details of the phonon mode calculations are in the Supplementary Information (Supplementary Information §9 and Supplementary Fig. 16). Using the calculated $S_{HT}(T)$ and $S_{LT}(T)$ curves under pressure and the SD model simulations, computational simulations on the adiabatic temperature change at $p = 280$ MPa were performed. According to the calculated $S_{HT}(T)$ and $S_{LT}(T)$ curves, the HT phase is transformed to the LT phase via isothermal pressurization with $\Delta S_{rev,calc}$ (calculated reversible entropy change) of $-169$ J K$^{-1}$ kg$^{-1}$ at 319 K. This agrees with the observed $\Delta S_{rev}$ value. This large $\Delta S_{rev,calc}$ value can be explained by the sum of two contributions. The first is the large difference in the phonon vibrational entropy ($\Delta S_{vib} = 122$ J K$^{-1}$ kg$^{-1}$) due to the electronic

states of Mn and Fe. The other is a large change in the spin–orbit entropy ($\Delta S_{os} = 47$ J K$^{-1}$ kg$^{-1}$). During the adiabatic pressure release process, the system should exhibit $|\Delta T_{ad,rev,calc}|$ (calculated reversible adiabatic temperature change) of 59 K (Fig. 4a, inset). This value agrees well with the $|\Delta T_{ad,rev}|$ value in Fig. 3.

From the viewpoint of raising the transition entropy, paramagnetic metal complexes such as molecular magnets and single-molecule magnets[50–53] should be suitable for barocaloric effect materials. This is due to their advantageous $S_{os}$.

We computationally calculated the refrigeration cycle for cyano-RbMnFeCo using first-principles phonon mode calculations and SD model simulations (Fig. 4b). The calculation involved three steps. (i) Apply pressure to the HT phase at an operation temperature to generate the LT phase through an isothermal pressurization process. As the material transforms from the HT phase to the LT phase, the system releases heat energy to a heat sink. (ii) Detach the heat sink at the same time as the pressure is released under an adiabatic depressurization condition. The LT phase returns to the HT phase. Additionally, this release is accompanied by a temperature decrease by $\Delta T_{ad,rev,calc}$ at constant entropy (rainbow arrow). Afterward, (iii) the system makes contact with the surroundings. The cold energy is transferred to the surroundings, and cyano-RbMnFeCo returns to the original HT phase. Cyano-RbMnFeCo cools the surroundings by repeating the pressure application and pressure release cycles (see Supplementary Movie 1).

### Direct measurement of the temperature change using a thermocouple upon applying and releasing the pressure

To practically measure the temperature change ($\Delta T_{obs}$) upon applying and releasing the pressure, we constructed an own-made apparatus using a thermocouple (non-adiabatic system) (Supplementary Fig. 17). In this system, a powder sample mixed with an organic binder is placed in the pressure cell, which is subsequently placed inside an incubator. Then a pump applies uniaxial pressure. A thermocouple is mounted inside the sample to measure the sample temperature. Figure 5a shows the results operated at 9 °C (282 K). Applying pressure (440 MPa) increases the temperature with $\Delta T_{obs} = +44$ K (9 °C → 53 °C), while releasing pressure decreases the temperature with $\Delta T_{obs} = -31$ K (9 °C → −22 °C). A remarkably large temperature change of 75 K (= +44 K + |−31| K) is detected in one cycle. The surrounding temperature and the temperature change are expressed in °C and K units, respectively.

Next, we conducted measurements at different operation temperatures (Supplementary Fig. 18). The temperature increase upon applying pressure is $\Delta T_{obs} = +28$ K, +39 K, +41 K, + 42 K, +44 K, +39 K, +35 K, +25 K, +17 K, +13 K, +10 K, and +10 K at starting temperatures (=operation temperatures) of −15 °C, −10 °C, −5 °C, 0 °C, 9 °C, 20 °C, 30 °C, 40 °C, 50 °C, 59 °C, 69 °C, and 77 °C, respectively. By contrast, the temperature decrease upon releasing pressure is $\Delta T_{obs} = -5$ K, −8 K, −13 K, −17 K, −31 K, −32 K, −35 K, −30 K, −28 K, −16 K, −12 K, and −10 K, respectively, at starting temperatures of −15 °C, −10 °C, −5 °C, 0 °C, 9 °C, 20 °C, 30 °C, 40 °C, 50 °C, 59 °C, 69 °C, and 77 °C, respectively. The decrease of the $\Delta T_{obs}$ value upon pressure release below 0 °C indicates that the operation temperature is approaching $T_\uparrow$ at 0.1 MPa, and decrease of $\Delta T_{obs}$ upon pressure application above 50 °C indicates that the operation temperature is approaching $T_\downarrow$ at 440 MPa.

Figure 5b plots the temperature increase and decrease at each starting temperature along the entropy curve under pressure in Fig. 3. The measurement results for different operation temperatures show that temperature ranges of 109 K between −22 °C and 87 °C within the $RC_{rev}$ region (light blue area) and 122 K from −32 °C to 90 °C were experimentally confirmed as the reversible barocaloric region. These results validate our barocaloric parameters and the magnitude of the values such as $|\Delta T_{ad,rev}|$. In cyano-RbMnFeCo, such a barocaloric effect is realized over a wide temperature range above and below room temperature because the phase transition temperature between the HT and LT phases are strategically adjusted by introducing Co ions.

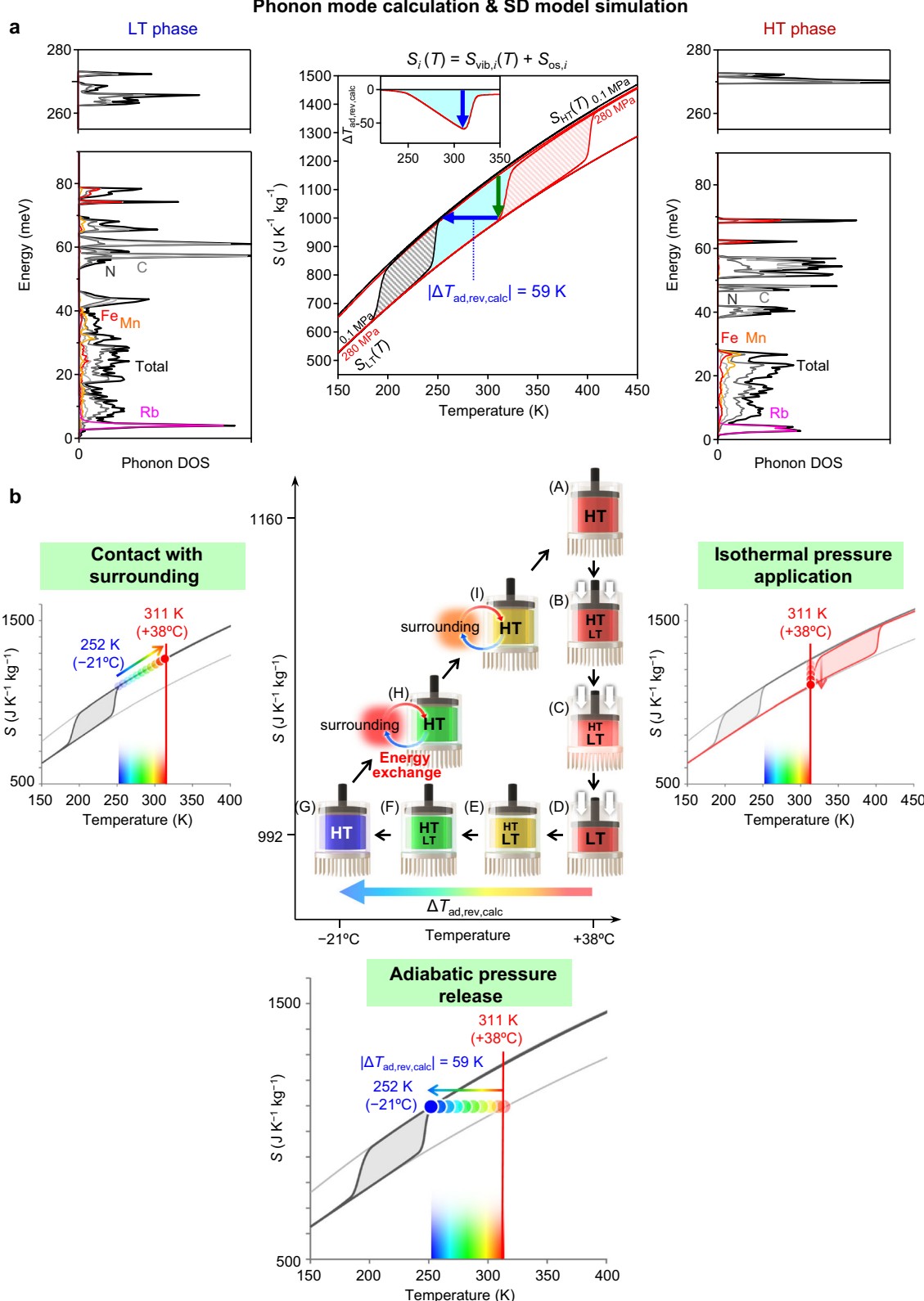

Another significant factor for solid-state refrigerants is cycle durability against applying and releasing pressure. We investigated repeated cycles using our direct measurement system. In this experiment, the average temperature change upon releasing pressure is $-30.5 \pm 0.2$ K for the first 3 cycles, while the average value for the last 7 cycles, (97th to 103rd cycle) is $-30.9 \pm 0.2$ K, indicating that 100 cycles of applying and releasing pressure do not affect the performance

(Fig. 5c). One reason that the present system does not decompose or easily break upon structural change during the phase transition is that the cyano-ligands connect the metal ions three-dimensionally.

## Discussion

**Cyano-RbMnFeCo** exhibits a giant reversible barocaloric effect with $|\Delta T_{ad,rev}|$ of 64 K, $T_{span,rev}$ of 102 K, and $RC_{rev}$ of 15600 J kg⁻¹ at

**Fig. 4 | Theoretical calculation of the reversible barocaloric effect in cyano-RbMnFeCo by first-principles phonon mode calculations and SD model simulations. a**, Phonon density of states (phonon DOS) of the LT phase (left) and the HT phase (right) for RbMn[Fe(CN)$_6$] obtained by first-principles phonon mode calculations considering thermal expansion. Center shows the calculated temperature dependences of the total entropy $S_i(T)$ ($i$ = HT or LT). Dependences are the sum of the vibrational entropy $S_{\text{vib},i}(T)$ and the spin–orbit entropy $S_{\text{os},i}$. Calculated entropy values for each temperature are interpolated to draw the entropy curves. Black and red thin lines indicate the entropy curves simulated by the SD model at 0.1 MPa and 280 MPa, respectively. Black and red shaded areas denote the thermal hysteresis loops at 0.1 MPa and 280 MPa, respectively. Light blue area denotes the reversible

barocaloric effect region. Green arrow indicates the isothermal pressurization process, and blue arrow indicates the adiabatic pressure release process showing the calculated reversible adiabatic temperature change ($|\Delta T_{\text{ad,rev,calc}}|$). Insets show the temperature dependences of $\Delta T_{\text{ad,rev,calc}}$. **b**, Schematic illustration of the reversible refrigeration cycle using cyano-RbMnFeCo. Cycle is composed of an isothermal pressure application process (A → B → C → D), an adiabatic pressure release process (D → E → F → G), and an energy exchange process via contact with the surroundings (G → H → I → A). Panels outside the cycle scheme are snapshots of the entropy curves for each process. Rainbow scales on the temperature axes represent the temperature of the cyano-RbMnFeCo solid refrigerant.

---

280 MPa. These increase to $|\Delta T_{\text{ad,rev}}|$ of 74 K, $T_{\text{span,rev}}$ of 112 K, and $RC_{\text{rev}}$ of 18700 J kg$^{-1}$ at 340 MPa, and furthermore, to $|\Delta T_{\text{ad,rev}}|$ of 85 K, $T_{\text{span,rev}}$ of 142 K, and $RC_{\text{rev}}$ of 26000 J kg$^{-1}$ at 560 MPa. The $|\Delta T_{\text{ad,rev}}|$, $T_{\text{span,rev}}$, and $RC_{\text{rev}}$ values are much larger than those previously reported for caloric effects in solid–solid phase transition refrigerants (Supplementary Tables 3, 5, and 6). The temperature change upon applying and releasing the pressure was experimentally evaluated using an own-made experimental setup to confirm the large barocaloric effect of the present material. A direct temperature change measurement using a thermocouple gave $\Delta T_{\text{obs}}$ = +44 K upon applying pressure. The temperature decrease upon releasing pressure was $\Delta T_{\text{obs}}$ = −31 K, resulting in the total temperature change of 75 K for one cycle. To date, such large $\Delta T_{\text{obs}}$ values have yet to be directly measured in solid–solid phase transition materials, although pressure-induced solid–liquid transitions in $n$-alkanes ($n$ = 16 and 18) have been reported to show a large temperature change of +57 K at 400 MPa[26]. In the durability experiment, the $\Delta T_{\text{obs}}$ value of cyano-RbMnFeCo did not degrade after 103 cycles of applying and releasing pressure.

The giant reversible barocaloric effect of the present material originates from four factors. (i) The temperature shift of the phase transition under pressure has a huge d$T$/d$p$ value. (ii) $\Delta S_{\text{rev}}$ originates from the contributions of the spin–orbit entropy $S_{\text{os}}$ along with the vibrational entropy $S_{\text{vib}}$ accompanying the charge transfer. Considering the pressure responsivity in (i), the barocaloric strength of the present material is $\Delta S_{\text{rev}}/\Delta p$ = 2.0 J K$^{-1}$ kg$^{-1}$ MPa$^{-1}$ at 50 MPa. (iii) The entropy curves of the LT and HT phases have gentle slopes around room temperature. This generates a large temperature shift during the adiabatic cooling or heating process at constant entropy. In the case of cyano-RbMnFeCo, the slope of the entropy versus temperature curve is 2.3 J K$^{-2}$ kg$^{-1}$, which is much smaller than 5.6 J K$^{-2}$ kg$^{-1}$ of neopentyl glycol (NPG)[54] or 7.1 J K$^{-2}$ kg$^{-1}$ of neopentyl alcohol (NPA)[55]. This results in a large $|\Delta T_{\text{ad,rev}}|$ value, which is about 2.5 times that of NPG or NPA. Furthermore, (iv) cyano-RbMnFeCo possesses a high heat durability up to 533 K (260 °C) along with a chemical stability against heat-transfer media.

From the viewpoint of practical applications, (i) a high thermal conductivity, (ii) a low necessary pressure, and (iii) a wide temperature window crossing room temperature are vital to develop caloric devices. For (i), the present material series exhibits a high thermal conductivity of $\lambda$ = 20 W m$^{-1}$ K$^{-1}$. Such a high $\lambda$ value is an advantage because it is suitable for high-frequency operations of barocaloric devices. For (ii), the present material shows $|\Delta T_{\text{ad,rev}}|$ = 21 K and $RC_{\text{rev}}$ = 3700 J kg$^{-1}$ at $p$ = 90 MPa (0.9 kbar). For (iii), the wide temperature window of $T_{\text{span,rev}}$ = 126 K suggests that the cascade method, which combines multiple refrigerants with different temperature ranges in a stepwise manner for heating and cooling, is unnecessary. Thus, cooling and heating can generate a large temperature difference in one step. For example, the present material may realize instant cooling from 100 °C to 25 °C or instant freezing from 25 °C to −50 °C. Additionally, the material cost for mass production is reasonable (Supplementary Information §11).

Tuning the transition temperature is one of the strategies to efficiently pump heat. Controlling the Co content is an effective approach to fully utilize the large $RC_{\text{rev}}$ value in the present material since the edge of the barocaloric window can be adjusted so that it matches the operation temperature. From the viewpoint of device applications, attaching the present material to a piezoelectric substrate may realize a compact solid refrigerant, which can prevent overheating in devices. The present work should open new possibilities in the field of barocaloric effect materials and contribute to the development of new refrigerants suitable for air conditioners and food storage.

## Methods
### Materials and characterization
Figure 1a depicts the procedure to prepare the target compound. A mixed aqueous solution of manganese(II) chloride (0.1 mol dm$^{-3}$) and rubidium chloride (1.0 mol dm$^{-3}$) was reacted with a mixed aqueous solution of potassium hexacyanoferrate (III) (0.1 mol dm$^{-3}$), potassium hexacyanocobaltate (III) (7 mmol dm$^{-3}$), and rubidium chloride (1.0 mol dm$^{-3}$). The solution was stirred at 50 °C. The precipitate was filtered and dried, yielding a powder sample. Then the sample was cooled by liquid nitrogen and warmed to room temperature three times. Manganese chloride tetrahydrate, rubidium chloride, and potassium hexacyanoferrate were purchased from FUJIFILM Wako, and potassium hexacyanocobaltate was purchased from Sigma-Aldrich. All reagents were used as received. Elemental analyses were performed using a standard microanalytical method and an inductively coupled plasma mass spectrometer (ICP-MS, Agilent 7700x). The sample had a formula of RbMn{[Fe(CN)$_6$]$_{0.92}$[Co(CN)$_6$]$_{0.08}$}·0.3H$_2$O: Calculated; Rb, 23.9; Mn, 15.4; Fe, 14.4; Co, 1.3; C, 20.1; N, 23.5; H, 0.2%: Found; Rb, 23.9; Mn, 15.3; Fe, 14.2; Co, 1.2; C, 20.1; N, 23.6; H, 0.2%.

The PXRD measurements were conducted using a RIGAKU Ultima IV with Cu Kα radiation (1.5418 Å), and the crystal structures were determined by Rietveld analyses of the PXRD patterns using RIGAKU PDXL software. In the Rietveld analysis, the positions and occupancies of water molecules were placed based on their positions reported in the single-crystal structure analyses and the compositional analysis of an analogue rubidium–manganese hexacyanoferrate system, Rb$_{0.61}$Mn[Fe(CN)$_6$]$_{0.87}$·1.7H$_2$O. The composition of the obtained material was RbMn{[Fe(CN)$_6$]$_{0.92}$[Co(CN)$_6$]$_{0.08}$}·0.3H$_2$O, and 0.3H$_2$O existed at the interstitial sites as non-coordinated water molecules. CCDC-2097442 and CCDC-2097443 contain the structural information of the HT and LT phases, respectively, in a crystallographic information file format and are available from the Cambridge Crystallographic Data Centre via https://www.ccdc.cam.ac.uk/structures/. The densities of the HT and LT phases were 2.00 g cm$^{-3}$ and 2.23 g cm$^{-3}$, respectively.

DSC measurements were carried out in a nitrogen atmosphere using a RIGAKU DSC8230 at a scan rate of 10 °C min$^{-1}$. The powder sample (10.93 mg) and the reference Al$_2$O$_3$ powder (10.05 mg) were crimped into an aluminum cell for the measurement. Heat capacity measurements were conducted by the relaxation method using a Quantum Design 6000 PPMS. The powder sample was pressed into a pellet for the measurement (5.64 mg and 5.68 mg for the LT phase and

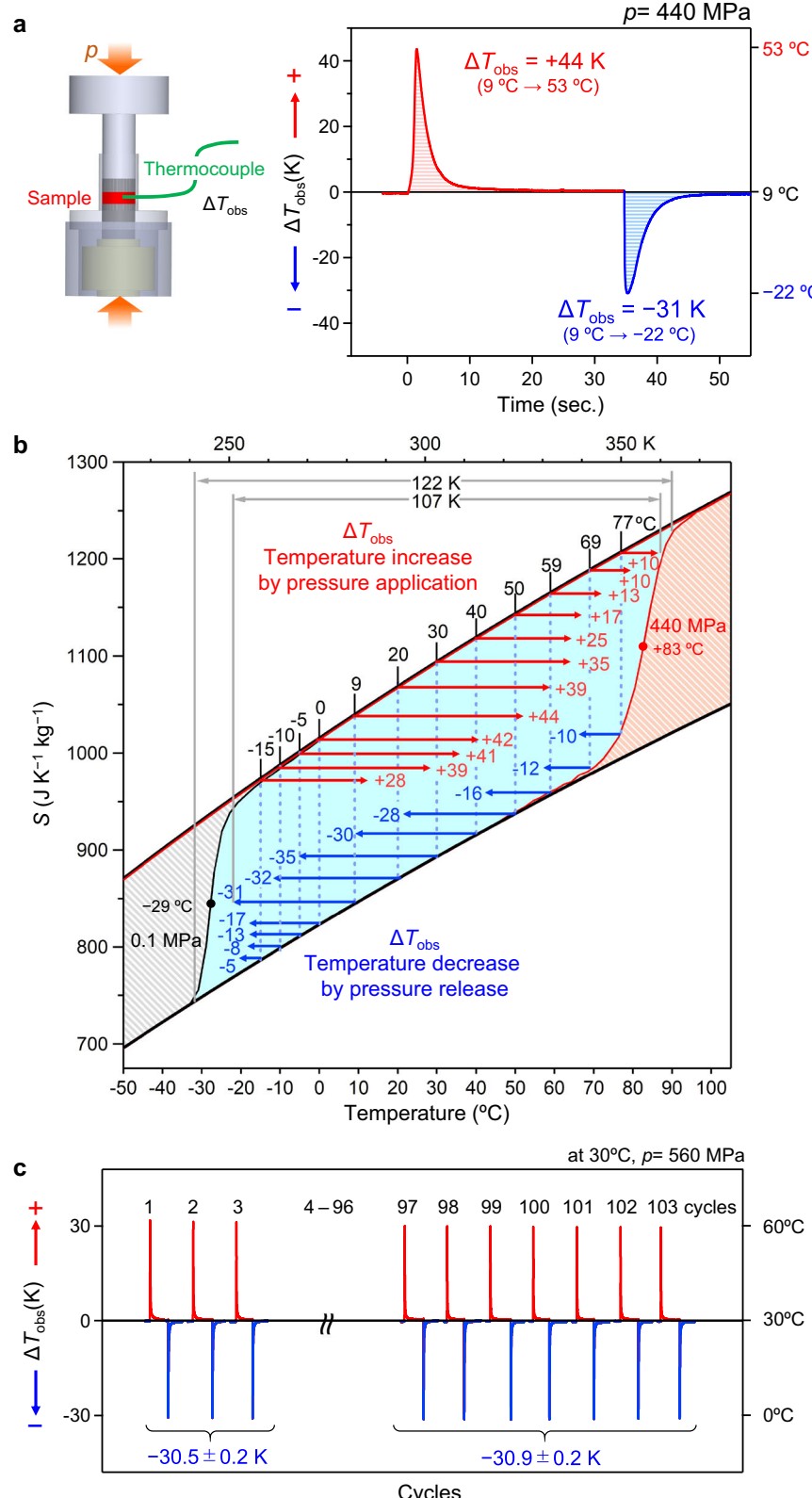

**Fig. 5 | Direct measurement of temperature change (ΔTobs) upon applying and releasing pressure in the barocaloric effect of cyano-RbMnFeCo. a,** Left shows the illustration of our experimental setup using a thermocouple. Right shows the observed temperature change ($\Delta T_{obs}$) upon applying pressure (red) and releasing pressure (blue) measured at 9 °C (282 K). Pressure was 440 MPa. **b,** Mapping of $\Delta T_{obs}$ on the entropy versus temperature curves. Black and orange shaded areas indicate the thermal hysteresis loops at 0.1 MPa and 440 MPa, respectively. Light blue area denotes the reversible barocaloric effect region. Red arrows indicate $\Delta T_{obs}$ upon applying pressure, and blue arrows indicate $\Delta T_{obs}$ upon releasing pressure for each starting temperature. **c,** Cycle durability against applying and releasing pressure of cyano-RbMnFeCo. The upper numbers indicate the number of cycles. Blue values show the average temperature decrease upon releasing the pressure for (left) the first 3 cycles (1st–3rd) and (right) the last 7 cycles (97th–103rd) at 560 MPa.

HT phase measurements, respectively). The amount of heat given at each measurement point (modulation amplitude) was set to 0.5% of the measurement temperature. Details of the devices are shown in Supplementary Table 7.

## Thermal conductivity measurement

The thermal conductivity was measured using a cyano-RbMnFe crystal. The thermal conductivity values were converted from the thermal effusivity acquired by a thermoreflectance technique using a thermal microscope TM3B (Bethel Co., Ltd.)[56]. The crystal sample was coated by a thin layer of Mo. Then the surface was periodically heated by an 808-nm diode laser, and the phase shift of the temperature response on the sample surface was detected by a 633-nm diode laser to obtain the thermal effusivity. Calibration was conducted using Ge, Si, and Pyrex glass as references.

## Measurement of the $\chi_M T$–$T$ plots under pressure

The magnetic properties under pressure were measured using a Cu-Be piston–cylinder clamp cell (Electrolab) in the SQUID magnetometer (Quantum Design, MPMS 7). A Teflon tube inside the cylinder was filled with the powder sample and Daphne 7373 as the pressure medium. Then the cylinder was sealed with a Teflon lid. The background signals were extracted to obtain the magnetic susceptibility data from the material itself. The applied pressures were read from the pressure measurement film (FUJIFILM) for pressures below 100 MPa, while for pressures above 100 MPa, the applied pressures were determined from the superconducting transition temperature of the lead set inside the cell.

## Fitting to the $\chi_M T$–$T$ plots

Based on the MF theory, the $\chi_M T$ value of the HT phase was fitted using a two-component MF model containing $Mn^{II}$ ($^6A_{1g}$, $S = 5/2$, $g = 2$) and $Fe^{III}$ ($^2T_{2g}$, $S = 1/2$, $g = 2$) with ferromagnetic ($0.5\,cm^{-1}$) and antiferromagnetic ($-2.0\,cm^{-1}$) superexchange interactions. By contrast, that of the LT phase was fitted with an MF model containing $Mn^{III}$ ($^5B_{1g}$, $S = 2$, $g = 2$) and ferromagnetic superexchange interaction ($0.5\,cm^{-1}$).

## Parameters of the SD model simulations

The parameters of the SD model simulations ($\Delta H$, $\Delta S$, $\gamma$) for each pressure were: ($13.1\,kJ\,mol^{-1}$, $58.0\,J\,K^{-1}\,mol^{-1}$, $6.8\,kJ\,mol^{-1}$) for 0.1 MPa, ($17.4\,kJ\,mol^{-1}$, $57.8\,J\,K^{-1}\,mol^{-1}$, $9.5\,kJ\,mol^{-1}$) for 50 MPa, ($18.7\,kJ\,mol^{-1}$, $57.6\,J\,K^{-1}\,mol^{-1}$, $10.1\,kJ\,mol^{-1}$) for 90 MPa, ($19.8\,kJ\,mol^{-1}$, $57.3\,J\,K^{-1}\,mol^{-1}$, $10.5\,kJ\,mol^{-1}$) for 130 MPa, ($20.5\,kJ\,mol^{-1}$, $57.0\,J\,K^{-1}\,mol^{-1}$, $10.7\,kJ\,mol^{-1}$) for 170 MPa, ($21.0\,kJ\,mol^{-1}$, $56.7\,J\,K^{-1}\,mol^{-1}$, $10.8\,kJ\,mol^{-1}$) for 210 MPa, ($21.4\,kJ\,mol^{-1}$, $56.3\,J\,K^{-1}\,mol^{-1}$, $10.8\,kJ\,mol^{-1}$) for 280 MPa ($21.6\,kJ\,mol^{-1}$, $55.9\,J\,K^{-1}\,mol^{-1}$, $10.9\,kJ\,mol^{-1}$) for 340 MPa, ($21.7\,kJ\,mol^{-1}$, $55.7\,J\,K^{-1}\,mol^{-1}$, $10.9\,kJ\,mol^{-1}$) for 380 MPa, and ($21.8\,kJ\,mol^{-1}$, $54.8\,J\,K^{-1}\,mol^{-1}$, $10.9\,kJ\,mol^{-1}$) for 560 MPa (Supplementary Figs. 9 and 10).

## Reporting summary

Further information on research design is available in the Nature Portfolio Reporting Summary linked to this article.

# Data availability

Data that support the findings of this study are presented in the main article and Supplementary Information files. Source data are provided in this paper. The X-ray crystallographic coordinates for the structure reported in this study have been deposited at the Cambridge Crystallographic Data Centre (CCDC), under deposition numbers 2097442 and 2097443. These data can be obtained free of charge from The Cambridge Crystallographic Data Centre via www.ccdc.cam.ac.uk/data_request/cif.

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

## Acknowledgements

This work was supported by a JSPS Grant-in-Aid for Scientific Research (A) (Grant Number 20H00369, S.O.). F.J. is grateful to JST SPRING (Grant Number JPMJSP2108). This work was also partially carried out in the framework of the IRL DYNACOM of CNRS (S.O.). We recognize the Cryogenic Research Center at The University of Tokyo, the Center for Nano Lithography & Analysis at The University of Tokyo, and Quantum Leap Flagship Program (Q-LEAP) supported by MEXT (S.O.). We are grateful to T. Tabata for his support with sample synthesis and M. Tanaka for her assistance with elemental analysis.

## Author contributions

S.O. coordinated this study, wrote the paper, and contributed to the measurements, analyses, and calculations. K.N. characterized the materials and conducted the thermodynamic and magnetic measurements. M.Y. contributed to the first-principles calculations and helped draft the paper. A.N. contributed to the thermodynamic analysis and helped prepare the figures. K.I. conducted thermodynamic measurements, magnetic measurements, and thermodynamic analyses. Y.N. contributed to the sample characterization. F.J. carried out the thermodynamic simulations. O.S. and J.W. conducted the literature review. H.T. performed the heat capacity measurements. T.S. carried out the high-pressure DSC measurements. K.C. implemented the first-principles phonon mode calculations. K.M., K.I., K.N., T.M. and R.H. prepared the samples and directly measured the temperature change upon applying and releasing the pressure.

## Competing interests

The authors declare no competing interests.
