## [Peer Review File · Nature Communications]

Giant adiabatic temperature change and its direct measurement of a barocaloric effect in a charge-transfer solidREVIEWER COMMENTS

Reviewer #1 (Remarks to the Author):

In abstract “ gaseous refrigerants negatively impact the environment and contribute to global warming” is not rigorous, the absolute certain tone used here is not appropriate.

There's redundancy in the abstract.

No photos showing sample size of 3.6+/- 1.9 microns

It's strange there's no abrupt xrd pattern change at around 192K and 248K

Explain the lab made heater and time it takes to transfer the pressure cell.

Neutron scattering experiments should be carried out measuring the phonon modes as a confirmation of the theory

The thermocouple measurement is interesting but could be problematic due to unknown/unexpected signals could come from uniaxial pressure. In order to prove its feasibility calibration experiment needs to be carried out to the thermocouple.

Reviewer #2 (Remarks to the Author):

The barocaloric effect has the potential to be utilized for environmentally friendly refrigeration. In this study, the authors present a novel inorganic barocaloric refrigerant, cyano-RbMnFeCo. Comprehensive experimental and theoretical investigations were conducted to assess the barocaloric performance of cyano-RbMnFeCo and elucidate its underlying mechanism. Notably, this compound exhibits several advantages compared to previously reported barocaloric materials, including a wide operating temperature range, relatively high thermal conductivity, and large barocaloric coefficient. Therefore, I recommend its publication in Nature Communications. However, certain aspects need to be addressed.

1. In the abstract, as well as on pages 9, 11, and 12, the authors assert that the adiabatic temperature change and the refrigerant capacity in cyano-RbMnFeCo are the largest among all caloric effects (or a word record). This claim does not hold true. For cyano-RbMnFeCo, an adiabatic temperature change of 44 K can be induced by a pressure of 440 MPa. In reference 26, a temperature change of 57 K is achieved with only 400 MPa for n-alkanes. Furthermore, in terms of refrigerant capacity, cyano-RbMnFeCo is not superior to n-alkanes. For instance, for cyano-RbMnFeCo, the refrigerant capacity is 26000 Jkg⁻¹ at 560 MPa. But for n-alkane (C₁₈H₃₈), due to the high reversible entropy change (about 700 Jkg⁻¹K⁻¹), only 232 MPa can lead to such a refrigerant capacity. So, the related sentences should be rewritten. Besides, the size of the applied pressure should be mentioned along with the values of directly measured adiabatic temperature and the refrigerant capacity.

2. On page 3, references 42-44 are cited to support the notion that metal-insulator transition (MIT) materials may exhibit a significant reversible barocaloric effect. However, these references do not pertain to the barocaloric effect specifically. Instead, Hexagonal nickel-iron

sulfide, a typical MIT compound, that has been reported to show giant barocaloric effect (Mater. Horiz., 2020, 7, 2690-2695).

3. Cyano-RbMnFeCo displays large thermal hysteresis which increases at higher pressures, which was reproduced by SD calculations on page 7. But, the underlying reason behind such remarkable changes in thermal hysteresis with pressure remains unclear. Therefore, further discussions are desired.

4. On page 10, a uniaxial pressure was utilized to measure the adiabatic temperature changes. However, the use of uniaxial pressure differs significantly from hydraulic pressure particularly when applied to inorganic compounds. The possible effects of these different pressure loadings on the adiabatic temperature should be discussed.

5. On page 11 and in Fig. S11, it is observed that as the starting temperature approaches -15°C, the application of pressure results in a significantly larger increase in temperature compared to the decrease in temperature upon releasing the same pressure. However, this asymmetric behavior is not evident when the starting temperature approaches 77°C. How to understand this?

6. On page 13, the comparison of thermal conductivity between cyano-RbMnFeCo and Al₂O₃ is not so meaningful because Al₂O₃ is not typically considered a high-thermal-conductivity material.

7. Page 13, it is claimed that the refrigerant volume of solid cyano-RbMnFeCo may be reduced to 1-2% compared with current gaseous refrigerants. Additional discussion and supporting data are necessary to substantiate this claim.

8. what does the asterisk in the XRD pattern (Fig. 1b and 1e) represent for?

Reviewer #3 (Remarks to the Author):

The authors present a study on the barocaloric effect in the Prussian blue analogue ((cyano-RbMnFeCo) showing the charge transfer process. The applications of molecular magnets in magnetic refrigeration were considered for many years, mainly in the sub-Kelvin temperatures range. The presented paper shows, that molecular magnetic materials can be also considered as a room- temperature coolers and can open a new perspective on molecular magnet applications.

In general, the presented paper shows excellent results, supported by the deep data analysis and what is most important, the direct measurements of the temperature change using a self-constructed system based on thermocouples. The work is clear and the outcome could be interesting for the community. In my opinion, this paper is suitable for publication in the Journal of Nature Communications.

A few minor comments should be addressed:

- The reason for choosing RbMn{[Fe(CN)₆]_{0.92}[Co(CN)₆]_{0.08}}·0.3H₂O instead of pure RbMnFe network is unclear. What is the advantage of Co ions in this system?
- The "Methods" section is incomplete- some of the techniques are not described while the paragraph materials cost is not necessary and should be sifted to SI or Introduction.
- The paper will be even better if include the study of pressure-dependent PXRD studies

Response to Reviewer 1

We greatly appreciate your constructive comments. Below is our response to each of your comments.

Comment 1. In abstract “gaseous refrigerants negatively impact the environment and contribute to global warming” is not rigorous, the absolute certain tone used here is not appropriate.

Answer 1. This sentence was written in the Introduction section. As per the reviewer’s comment, this sentence was removed.

Comment 2. There’s redundancy in the abstract.

Answer 2. The abstract is revised as follows:

Revised Abstract: Solid refrigerants exhibiting a caloric effect upon applying external stimuli **are receiving attention as one of the next-generation refrigeration technologies**. Herein, we report a new inorganic refrigerant, rubidium cyano-bridged manganese–iron–cobalt ternary metal assembly (**cyano-RbMnFeCo**). **Cyano-RbMnFeCo** shows a reversible barocaloric effect with large reversible adiabatic temperature changes ($|\Delta T_{\text{ad,rev}}|$) of 74 K (from 57 °C to –17 °C) at 340 MPa, and 85 K (from 88 °C to 3 °C) at 560 MPa. **Such large $|\Delta T_{\text{ad,rev}}|$ values have yet to be reported among caloric effects in solid-solid phase transition refrigerants**. The reversible refrigerant capacity (RC_{rev}) is 26000 J kg^{–1} and the temperature window ($T_{\text{span,rev}}$) is 142 K. Additionally, **cyano-RbMnFeCo** shows barocaloric effects even at low pressures, e.g., $|\Delta T_{\text{ad,rev}}| = 21$ K at 90 MPa. Furthermore, direct measurement of the temperature change (ΔT_{obs}) using a thermocouple shows $\Delta T_{\text{obs}} = +44$ K by applying pressure. The temperature increase and decrease upon pressure application and release are repeated over 100 cycles without any degradation of the ΔT_{obs} performance. This material series also possesses a high thermal conductivity value of 20.4 W m^{–1} K^{–1}. The present barocaloric material may realize a high-efficiency solid refrigerant.

Comment 3. No photos showing sample size of 3.6 ± 1.9 microns.

Answer 3. We added the SEM image and particle size distribution as Fig. S1 in the Supplementary Information.

Figure S1. SEM image and size distribution of **cyano-RbMnFeCo**.

Comment 4. It's strange there's no abrupt XRD pattern change at around 192 K and 248 K.

Answer 4. Thank you for this insightful comment. We thoroughly checked the conditions of the variable-temperature PXRD (VT-PXRD) measurement and found a discrepancy between the sample temperature and the temperature of the thermometer at the sample holder, which is attributed to an insufficient vacuum level inside the cryostat. In the revised manuscript, we remeasured the VT-PXRD data under a high vacuum level of 2.4×10^{-4} Pa (Fig. A1). The transition temperatures agreed well with those obtained from the magnetic measurements. In the revised manuscript, Fig. S3 in the Supplementary Information is updated with the remeasured data.

Figure A1. Temperature dependence of the HT phase fraction obtained from the VT-PXRD measurement.

Figure S3. Temperature dependence of the PXRD patterns of cyano-RbMnFeCo. Red and blue numbers correspond to the indices for the HT and LT phases, respectively.

Comment 5. Explain the lab made heater and time it takes to transfer the pressure cell.

Answer 5. Detailed explanation of the procedure for sample heating above 400 K is as follows. The pressure cell with **cyano-RbMnFeCo** (Fig. A2a) was put in a glass tube along with a thermometer. Then the cell was heated in an oil bath (Fig. A2b). Once the desired temperature was reached, the pressure cell was transferred from the lab-made heater to the SQUID device in 20 s (Fig. A2c). Figure A2d shows the decay of the temperature versus time plot during this transfer, which is expressed as $T = 433 + 39 \exp(-t/30)$. Here, t is time in seconds. For example, when the sample was heated to 473 K in the lab-made heater, the sample temperature after 20 s of transfer from the lab-made heater to the SQUID magnetometer was 453 K. Then the measurement started after the sample was cooled in the SQUID device to 400 K, which is the upper temperature limit of SQUID. Cooling took about 10 min. A detailed explanation is added to the revised Supplementary Information.

Figure A2

Figure A2. Details of the sample transfer from the lab-made heater to the SQUID magnetometer. **a**, Photograph and schematic illustration of the pressure cell containing the **cyano-RbMnFeCo** sample. **b**, Photograph of the lab-made heater. **c**, Photograph of placing the pressure cell into the SQUID magnetometer. **d**, Temperature versus time plot during the transfer of the pressure cell from the lab-made heater to the SQUID device. Plot is fitted with an exponential decay.

Supplementary Information, Figure S5 legend: For measurements at pressures of 280–560 MPa, the sample was heated above 400 K using a lab-made heater because the upper temperature limit of the SQUID magnetometer was 400 K. First, the pressure cell containing **cyano-RbMnFeCo** was put in a glass tube together with a thermometer. Then it was heated in an oil bath. Once the desired temperature was reached, the pressure cell was transferred from the lab-made heater to the SQUID device in 20 s. The decay of the temperature versus time plot during this transfer showed an exponential decrease of $T = 433 + 39 \exp(-t/30)$. Here, t is time in seconds. When the sample cooled down to 400 K in the SQUID magnetometer set to 400 K, the magnetic measurement started.

Comment 6. Neutron scattering experiments should be carried out measuring the phonon modes as a confirmation of the theory.

Answer 6. Although we contacted accelerator facilities to inquire about beam time to conduct a neutron scattering experiment, the waitlist to reserve the beam time is very long. In place of neutron scattering measurements, we show the comparison between the calculated phonon mode frequencies and the experimentally obtained resonance frequencies in the Raman and IR spectra to validate the phonon mode calculations (Fig. A3). The calculated and observed Raman spectra agree well. In the Raman spectra, the resonance peaks in the region of 2100–2300 cm^{-1} are assigned to the CN stretching modes, which slightly differ; the maximum difference between the calculation and the experiment is 1.8% for the LT phase and 2.6% for the HT phase. The far-IR spectra also agree well with the calculated spectra. Therefore, we consider the phonon mode calculation results to be valid for theoretical discussion. This explanation is added to the Supplementary Information as follows.

Supplementary Information, Page S26, Line 14: It should be noted that the calculated Raman and IR frequencies agree well with the experimental spectra, indicating that the phonon mode calculation results are reasonable.

Figure A3

Figure A3. Raman and Far-IR spectra of cyano-RbMnFe. **a**, Observed Raman spectrum of the LT phase. Pink shading indicates the peak attributed to the residual HT phase. **b**, Observed Raman spectrum of the HT phase. **c**, Calculated Raman spectrum of the LT phase. **d**, Calculated Raman spectrum of the HT phase. **e**, Observed far-IR spectrum of the LT phase at 100 K. **f**, Observed far-IR spectrum of the HT phase at 300 K. **g**, Calculated IR spectrum of the LT phase. **h**, Calculated IR spectrum of the HT phase.

Comment 7. The thermocouple measurement is interesting but could be problematic due to unknown/unexpected signals could come from uniaxial pressure. In order to prove its feasibility calibration experiment needs to be carried out to the thermocouple.

Answer 7. Thank you for the insightful comment. The calibration temperature in response to pressure is 0.101 ± 0.003 K per 100 MPa. In the revised manuscript, the ΔT_{obs} values from the direct measurement experiment were calibrated. This is mentioned in the revised Supplementary Information.

Supplementary Information, Page S8, Right column:

Calibration for the temperature deviation by pressure: 0.101 ± 0.003 K per 100 MPa

Response to Reviewer 2

We greatly appreciate your constructive comments and suggestions to improve our manuscript. Below are our responses to each of your comments and an outline of the revisions to our manuscript.

Comment 0. The barocaloric effect has the potential to be utilized for environmentally friendly refrigeration. In this study, the authors present a novel inorganic barocaloric refrigerant, **cyano-RbMnFeCo**. Comprehensive experimental and theoretical investigations were conducted to assess the barocaloric performance of **cyano-RbMnFeCo** and elucidate its underlying mechanism. Notably, this compound exhibits several advantages compared to previously reported barocaloric materials, including a wide operating temperature range, relatively high thermal conductivity, and large barocaloric coefficient. Therefore, I recommend its publication in Nature Communications. However, certain aspects need to be addressed.

Answer 0. We appreciate the reviewer's high evaluation of our work. Below are the answers to each of your comments.

Comment 1. In the abstract, as well as on pages 9, 11, and 12, the authors assert that the adiabatic temperature change and the refrigerant capacity in **cyano-RbMnFeCo** are the largest among all caloric effects (or a world record). This claim does not hold true. For **cyano-RbMnFeCo**, an adiabatic temperature change of 44 K can be induced by a pressure of 440 MPa. In reference 26, a temperature change of 57 K is achieved with only 400 MPa for *n*-alkanes. Furthermore, in terms of refrigerant capacity, **cyano-RbMnFeCo** is not superior to *n*-alkanes. For instance, for **cyano-RbMnFeCo**, the refrigerant capacity is 26000 J kg⁻¹ at 560 MPa. But for *n*-alkane (C₁₈H₃₈), due to the high reversible entropy change (about 700 J kg K⁻¹), only 232 MPa can lead to such a refrigerant capacity. So, the related sentences should be rewritten. Besides, the size of the applied pressure should be mentioned along with the values of directly measured adiabatic temperature and the refrigerant capacity.

Answer 1. As the reviewer has pointed out, *n*-alkane, which exhibits a solid–liquid phase transition, shows a good barocaloric performance. On the other hand, since the present study focuses on “solid refrigerants due to solid–solid phase transition”, which does not accompany pressure-induced melting, we have revised the manuscript to make this point clear to the readers. As for the reviewer's additional comment on the applied pressure in the direct measurement on the temperature change, we added the information in Figure 5 of the revised manuscript. The corrections in the revised manuscript are as follows:

Abstract, Page 2, Line 6: Such large $|\Delta T_{\text{ad,rev}}|$ values have yet to be reported among caloric effects in solid–solid phase transition refrigerants.

Page 4, Line 4: These $|\Delta T_{\text{ad,rev}}|$ values are the largest reversible adiabatic temperature changes among caloric effects in solid–solid phase transition materials reported to date.

Page 9, Line 3: The $|\Delta T_{\text{ad,rev}}|$ values of 59–85 K are the largest ones among the caloric effects in solid–solid phase transition refrigerants.

Page 12, Line 5: The $|\Delta T_{ad,rev}|$, $T_{span,rev}$, and RC_{rev} values are much larger than those previously reported for caloric effects in solid–solid phase transition refrigerants (Tables S3, S5, and S6).

Page 12, Line 11: To date, such large ΔT_{obs} values have yet to be directly measured in solid–solid phase transition materials, although pressure-induced solid–liquid transitions in n -alkanes ($n = 16$ and 18) have been reported to show a large temperature change of $+57$ K at 400 MPa²⁶.

Comment 2. On page 3, references 42-44 are cited to support the notion that metal-insulator transition (MIT) materials may exhibit a significant reversible barocaloric effect. However, these references do not pertain to the barocaloric effect specifically. Instead, Hexagonal nickel-iron sulfide, a typical MIT compound, that has been reported to show giant barocaloric effect (*Mater. Horiz.*, 2020, 7, 2690-2695).

Answer 2. Thank you for your suggestion. The revised manuscript cites the literature on hexagonal nickel–iron sulfide as a typical MIT compound displaying a barocaloric effect.

References:

Ref. 42. Lin, J. et al. Giant room-temperature barocaloric effect at the electronic phase transition in $Ni_{1-x}Fe_xS$. *Mater. Horiz.* 7, 2690–2695 (2020).

Comment 3. Cyano-RbMnFeCo displays large thermal hysteresis which increases at higher pressures, which was reproduced by SD calculations on page 7. But, the underlying reason behind such remarkable changes in thermal hysteresis with pressure remains unclear. Therefore, further discussions are desired.

Answer 3. To understand the influence of the applied pressure on the changes in the thermal hysteresis, we discuss using the interaction parameter γ in the Slichter–Drickamer (SD) model. The γ parameter is interpreted as the interface stress between the HT and LT domains inside the crystal (Fig. A4).

Figure A4. Schematic illustration of the phase transition process with internal stress within the crystal.

The γ value of the SD model is known to depend on both the temperature and the pressure [ref. 48]. The temperature-dependent term $\gamma(T)$ is assumed here as $\gamma(T) = \gamma_{T,0} + \gamma_{T,1}\exp(aT)$. The pressure-dependent term $\gamma(p)$ is expressed as $\gamma(p) = \gamma_{p,1}p + \gamma_{p,2}p^2$. Thus, γ is $\gamma(T, p) = \gamma_{T,0} + \gamma_{T,1}\exp(aT) + \gamma_{p,1}p + \gamma_{p,2}p^2$. The plot of the γ values from the SD model simulations is well reproduced by the parameters of $\gamma_{T,0} = 10.7(1)$ kJ mol⁻¹, $\gamma_{T,1} = -1.2(5)\times 10^2$ kJ mol⁻¹, $a = -1.5(2)\times 10^{-2}$ K⁻¹, $\gamma_{p,1} = 2.9(5)\times 10^{-3}$ kJ mol⁻¹ MPa⁻¹, and $\gamma_{p,2} = -4.0(9)\times 10^{-6}$ kJ mol⁻¹ MPa⁻² (Fig. S8).

Figure S8. Pressure and temperature dependence of the interaction parameter (γ). Red circles show the interaction parameters analyzed from the SD model simulations. Black line is the reproduced curve using $\gamma(T, p) = \gamma_{T,0} + \gamma_{T,1} \exp(aT) + \gamma_{p,1} p + \gamma_{p,2} p^2$.

One of the reasons for the changes of the thermal hysteresis is considered as follows. The transition temperature shifts to higher temperatures upon applying external pressure, increasing the volume difference between the HT and LT phases. The increase in the volume difference induces a larger mismatch between the HT and LT phase domains, which is expected to enhance the surface stress at the domain interface. When 560 MPa pressure is applied, the volume difference should increase by 1.5% compared to that under atmospheric pressure due to the increased transition temperature.

The γ value plot and the explanation are added to the Supplementary Information as Figure S8.

Supplementary Information, Figure S8 legend: Pressure and temperature dependence of the interaction parameter (γ). The γ parameter in the SD model is interpreted as the interface stress between the HT and LT phase domains inside the crystal. The γ value is known to depend on both the temperature and the pressure⁴⁸. The temperature-dependent term $\gamma(T)$ is assumed here as $\gamma(T) = \gamma_{T,0} + \gamma_{T,1}\exp(aT)$. The pressure-dependent term $\gamma(p)$ is expressed as $\gamma(p) = \gamma_{p,1}p + \gamma_{p,2}p^2$. Thus, γ is $\gamma(T, p) = \gamma_{T,0} + \gamma_{T,1}\exp(aT) + \gamma_{p,1}p + \gamma_{p,2}p^2$. The plot of the γ values from the SD model simulations (red circles) is well reproduced by the parameters of $\gamma_{T,0} = 10.7(1)$ kJ mol⁻¹, $\gamma_{T,1} = -1.2(5)\times 10^2$ kJ mol⁻¹, $a = -1.5(2)\times 10^{-2}$ K⁻¹, $\gamma_{p,1} = 2.9(5)\times 10^{-3}$ kJ mol⁻¹ MPa⁻¹, and $\gamma_{p,2} = -4.0(9)\times 10^{-6}$ kJ mol⁻¹ MPa⁻² (black line). One of the reasons for the changes of the thermal hysteresis is considered as follows. The transition

temperature shifts to higher temperatures upon applying external pressure, increasing the volume difference between the HT and LT phases. The increase in the volume difference induces a larger mismatch between the HT and LT phase domains, which is expected to enhance the surface stress at the domain interface.

Comment 4. On page 10, a uniaxial pressure was utilized to measure the adiabatic temperature changes. However, the use of uniaxial pressure differs significantly from hydraulic pressure particularly when applied to inorganic compounds. The possible effects of these different pressure loadings on the adiabatic temperature should be discussed.

Answer 4. Because the current measurement is performed using randomly oriented powder samples, and the pressure is applied to the crystal from all directions, the present experiment is considered to be applying a pseudo-hydrostatic pressure. In the case of **cyano-RbMnFeCo** with a cubic crystal structure, an anisotropic barocaloric effect is expected to occur between the [100], [110], and [111] directions. For example, in the field of superconducting materials, superconducting properties are tuned by changing the direction of the uniaxial pressure. If a large single crystal of the present material can be prepared, then the barocaloric effect may be activated at lower pressures by selecting a specific crystallographic direction for the uniaxial pressure. This information is added to the main text in the revised manuscript.

Page 13, Line 16: Furthermore, if a large single crystal is realized in the future, the barocaloric effect may be activated at lower pressures by selecting a specific crystallographic direction for the pressure application.

Supplementary Information, Figure S12 legend: The pellet is composed of randomly oriented crystals.

Comment 5. On page 11 and in Fig. S11, it is observed that as the starting temperature approaches -15°C , the application of pressure results in a significantly larger increase in temperature compared to the decrease in temperature upon releasing the same pressure. However, this asymmetric behavior is not evident when the starting temperature approaches 77°C . How to understand this?

Answer 5. We appreciate this insightful comment. We performed an additional experiment at a different pressure (490 MPa). Results of ΔT_{obs} at $p = 490$ MPa shows an asymmetric temperature change around 76°C (Fig. A5). Therefore, the asymmetric behavior is not lost at high temperatures. The ΔT_{obs} value upon pressure release decreases at high temperatures due to the non-adiabatic environment of the present experimental setup using the thermocouple. Explanation is added to the revised Supplementary Information as below.

Supplementary Information, Figure S13 legend: The time decay of ΔT_{obs} depends on the starting temperature. In the future, we plan to improve the experimental setup for higher thermal insulation.

Figure A5. Direct measurement of the temperature change using a thermocouple. Mapping of ΔT_{obs} on the entropy versus temperature curves. Black and orange shaded areas indicate the thermal hysteresis loops at 0.1 MPa and 490 MPa, respectively. Red and blue arrows indicate ΔT_{obs} upon applying and releasing pressure, respectively, for each starting temperature.

Comment 6. On page 13, the comparison of thermal conductivity between **cyano-RbMnFeCo** and Al_2O_3 is not so meaningful because Al_2O_3 is not typically considered a high-thermal-conductivity material.

Answer 6. As per the reviewer's comment, we removed the comparison of the thermal conductivity with Al_2O_3 .

Comment 7. Page 13, it is claimed that the refrigerant volume of solid **cyano-RbMnFeCo** may be reduced to 1-2 % compared with current gaseous refrigerants. Additional discussion and supporting data are necessary to substantiate this claim.

Answer 7. Thank you for this insightful comment. The value of 1–2% was estimated from the comparison of the transition enthalpy per unit volume between the present material and gaseous refrigerants under atmospheric pressure. Since this comparison was inappropriate, the sentence was deleted from the manuscript.

Comment 8. What does the asterisk in the XRD pattern (Fig. 1b and 1e) represent for?

Answer 8. The asterisks in the PXRD patterns denote the peaks from the silicon standard used to calibrate the diffraction line positions. This information is added to the figure legend in the revised manuscript.

Figure 1b, 1e legend: Asterisk indicates the peak from the silicon standard.

Response to Reviewer 3

We greatly appreciate your constructive comments and suggestions to improve our manuscript. Below are our responses to each of your comments and an outline of the revisions to our manuscript.

Comment 0. The authors present a study on the barocaloric effect in the Prussian blue analogue (**cyano-RbMnFeCo**) showing the charge transfer process. The applications of molecular magnets in magnetic refrigeration were considered for many years, mainly in the sub-Kelvin temperatures range. The presented paper shows, that molecular magnetic materials can be also considered as a room- temperature coolers and can open a new perspective on molecular magnet applications.

In general, the presented paper shows excellent results, supported by the deep data analysis and what is most important, the direct measurements of the temperature change using a self-constructed system based on thermocouples. The work is clear and the outcome could be interesting for the community. In my opinion, this paper is suitable for publication in the Journal of Nature Communications.

A few minor comments should be addressed:

Answer 0. We appreciate the reviewer's high evaluation of our work. Below are the answers to each of your comments.

Comment 1. The reason for choosing $\text{RbMn}\{[\text{Fe}(\text{CN})_6]_{0.92}[\text{Co}(\text{CN})_6]_{0.08}\} \cdot 0.3\text{H}_2\text{O}$ instead of pure RbMnFe network is unclear. What is the advantage of Co ions in this system?

Answer 1. The T_{\uparrow} value of the phase transition from the LT phase to the HT phase in pure RbMnFe network is 304 K (31 °C), which is above room temperature. Unfortunately, a system using a pure RbMnFe system network cannot be cooled below room temperature. To realize a barocaloric effect over a wide temperature range around room temperature, materials with lower phase transition temperatures are necessary. In the present work, we found that partially replacing $[\text{Fe}(\text{CN})_6]$ with $[\text{Co}(\text{CN})_6]$ can lower the phase transition temperature. Therefore, we used the Co-substituted sample in the present work. The revised manuscript contains this explanation.

Page 11, Line 17: In **cyano-RbMnFeCo**, such a barocaloric effect is realized over a wide temperature range above and below room temperature because the phase transition temperature between the HT and LT phases are strategically adjusted by introducing Co ions.

Comment 2. The "Methods" section is incomplete- some of the techniques are not described while the paragraph materials cost is not necessary and should be shifted to SI or Introduction.

Answer 2. Thank you for this insightful comment. We revised the Methods section to include the technical details. The explanation of the material cost was moved from the Methods section to the Supplementary Information.

Methods, Page 14, Line 10: Elemental analyses were performed using a standard microanalytical method and an inductively coupled plasma mass spectrometer (ICP-MS, Agilent 7700x).

Methods, Page 14, Line 14: The PXRD measurements were conducted using a RIGAKU Ultima IV with Cu K α radiation (1.5418 Å), ...

Methods, Page 15, Line 12: The magnetic properties under pressure were measured using a Cu-Be piston–cylinder clamp cell (Electrolab) in the SQUID magnetometer (Quantum Design, MPMS 7).

Comment 3. The paper will be even better if include the study of pressure-dependent PXRD studies.

Answer 3. We contacted synchrotron radiation facilities to inquire about pressure-dependent PXRD measurements. However, it will take a long time to reserve for the beam time and to obtain the results.

We would like to consider the pressure-dependent measurement as a topic for future research.

REVIEWERS' COMMENTS

Reviewer #1 (Remarks to the Author):

most of my concerns in previous review are addressed, my only comment would be checking some of the results using synchrotron and neutrons, the long waiting time is not a very good excuse for such studies, but overall this manuscript worths be published on nature comm.

Reviewer #2 (Remarks to the Author):

The authors have addressed most of my concerns. However, one more modification is needed. In the response to my comment 4, the authors added a sentence in the revised manuscript about the barocaloric effect of single crystal sample. This is misleading because hydraulic pressure means pressing the sample from all directions. So, none direction can be selected to apply hydraulic pressure even if a single crystal is available. It is better to remove that sentence.

Reviewer #3 (Remarks to the Author):

The manuscript's authors have revised their contribution and replied to the reviewers' comments. Now, in my opinion, the manuscript is suitable for publication in Nature Communications.

Response to Reviewer 1

We greatly appreciate your constructive comments. Below is our response to your comment.

Comment 1. Most of my concerns in previous review are addressed, my only comment would be checking some of the results using synchrotron and neutrons, the long waiting time is not a very good excuse for such studies, but overall this manuscript worths be published on nature comm.

Answer 1. We greatly appreciate your comments and your evaluation. We are now making a plan for synchrotron and neutron measurements in the future.

Response to Reviewer 2

We greatly appreciate your constructive comments. Below is our response to your comment.

Comment 1. The authors have addressed most of my concerns. However, one more modification is needed. In the response to my comment 4, the authors added a sentence in the revised manuscript about the barocaloric effect of single crystal sample. This is misleading because hydraulic pressure means pressing the sample from all directions. So, none direction can be selected to apply hydraulic pressure even if a single crystal is available. It is better to remove that sentence.

Answer 1. We agree that hydraulic pressure is different from applying uniaxial pressure to randomly oriented powder sample. According to the reviewer's suggestion, we removed the corresponding sentence (Page 13, Line 16 in the previous manuscript).

Response to Reviewer 3

We greatly appreciate your constructive comments. Below is our response to your comment.

Comment 1. The manuscript's authors have revised their contribution and replied to the reviewers' comments. Now, in my opinion, the manuscript is suitable for publication in Nature Communications.

Answer 1. We greatly appreciate your comments and your evaluation.